# The Phylogenetic Significance of Fruit Structures in the Family Cornaceae of China and Related Taxa

**DOI:** 10.3390/plants11192591

**Published:** 2022-09-30

**Authors:** Jingru Wang, Hui Zou, Mei Liu, Yuting Wang, Jian Ru, Changhong Guo

**Affiliations:** 1Key Laboratory of Molecular Cytogenetic and Genetic Breeding of Heilongjiang Province, Department of Life Sciences, College of Life Science and Technology, Harbin Normal University, Harbin 150025, China; 2Department of Life Sciences, College of Life Science and Technology, Shangrao Normal University, Shangrao 334001, China

**Keywords:** Cornaceae, fruit, structure, morphology, systematics

## Abstract

The fruit morphological structures of the Cornaceae of China and related taxa were studied using the wax GMA semi-thin section method and other methods to identify characters useful in delimiting clades circumscribed in previous molecular phylogenetic studies. Maximum parsimony analyses of 27 fruit structural characters resulted in a generally poorly resolved strict consensus tree, yet one whose major clades matched those revealed previously. Cornaceae of China and related taxa are recognized in four significant clades with the following fruit structural features: (1) *Helwingia*, fruits lack trichome, the abdominal vascular bundles are close to the endocarp, and the endocarp sclereid is elongated; (2) *Aucuba*, single-cell lanceolate trichomes, pericarp without secretory structure; (3) *Torricellia*, polygon and elongated sclereids in the endocarp, pericarp without crystal and tannin; and (4) *Cornus* sensu lato, the trichome is T-shaped, the abdominal ventral bundle is absent, and the endocarp sclereid is nearly round. In *Cornus* sensu lato, this document supported that the cornelian cherries (CC, subg. *Cornus*) and the big-bracted dogwoods (BB, subg. *Syncarpea*) are sister groups. The dwarf dogwoods (DW, subg. *Arctocrania*) are sister to them, and the blue- or white-fruited dogwoods (BW, subg. *Kraniopsis*, subg. *Yinquania*, and subg. *Mesomora*) are the base of the *Cornus* sensu lato clade. The number of cell layers of endocarps and the types of crystals afford sound evidence for identifying their relationship. This study indicated that the fruit structures of Cornaceae might provide morphological and anatomical evidence for molecular phylogeny.

## 1. Introduction

The Cornaceae and related taxa are ecologically and horticulturally significant families, primarily trees or shrubs, but rarely herbs, which are found in the northern hemisphere’s tropical, temperate, and circumpolar regions [1]. Harms (1898) and Hutchinson (1967) positioned Cornaceae in the order Apiales based on its umbel or compound umbel, 2–5 carpels, inferior ovary, and one anatropous ovule per locule [2,3]. However, some scholars shifted their classification from Apiales to Cornales based on the majority of ligneous plants, simple leaves opposite (rarely alternate), drupes (rarely berry), and cyclic enol ether terpene compounds [4,5]. Cornaceae is one of the most complex families of flowering plants due to the highly divergent and plastic characteristics of its members. Harms (1898) classified 15 genera (*Alangium*, *Aucuba*, *Cornus*, *Helwingia*, and *Torricellia et al*.) predicated on their wood, bract, involucre, and fruit sclereid characteristics [2]. Eyde (1988) moved *Aucuba*, *Helwingia*, and *Torricellia* from Cornaceae and retained *Cornus*, *Camptotheca*, *Diplopanax*, *Davidia*, *Mastixia*, and *Nyssa* based on the characteristics of their germination valves, the chromosome number of x = 11, and iridoid compounds [6]. Fang and Hu (1990) indicated that Cornaceae contains *Cornus* sensu lato (*Bothrocaryum*, *Chamaepericlymenum*, *Cornus*, *Dendrobenthamia*, *Swida*), *Mastixia*, *Aucuba*, *Helwingia*, and *Torricellia*, according to leaves and inflorescence traits [1]. Xiang and Boufford (2005) demonstrated that Cornaceae solely contains *Cornus* sensu lato. [7].

Several genera (*Aucuba*, *Helwingia*, and *Torricellia* et al.) previously classified as Cornaceae were moved to new families based on other features by different researchers. *Helwingia* was classified into Cornaceae because of its umbel, lower ovary, and one ovule per cell [2,5,8,9,10,11,12,13,14,15]; into Araliaceae based on shrubs, alternate leaves, berry drupes [3,11,16,17,18,19]; or as Helwingiaceae according to the characteristics of the flowers, embryos, and pollen [4,20,21,22,23,24,25,26,27,28,29]. *Aucuba* was placed in Cornaceae according to its inferior ovary and one pendulous ovule per cell [2,3,5,9,15]; in Garryaceae based on flowers unisexual, leaves opposite, panicle or raceme, and chemical components [30,31]; or as a separate family Aucubaceae [32,33]. *Torricellia* was placed in Cornaceae according to the characteristics of the small arbour, lower ovary, 3–4 locules, and one pendulous ovule per cell [2,3,5,9,21]. It is also classified as a separate family, Torricelliaceae, because of its palmately lobed leaves, heteromorphic flowers, conical racemes, and pollen features [26,30,34].

Recent molecular phylogenetic studies have addressed genus relationships within Cornaceae and related taxa. The chloroplast gene rbcL sequence analysis revealed that *Aucuba*, *Helwingia*, and *Torricellia* are distantly related to Cornaceae and should not belong to Cornaceae [35,36,37]. Ngondya (2013) indicated that *Aucuba* has a distant relationship to the other species of Cornaceae, according to cpDNA and PCR-RFLP of rDNA data analysis [38]. Some researchers pointed to Cornaceae consisting of only *Alangium* and *Cornus* sensu lato [39,40,41]. Fan and Xiang (2003) asserted that *Cornus* sensu lato and *Alangium* are monophyletic taxa respectively based on 26S rDNA-matK-rbcL sequences analysis [42]. It proposed considering *Cornus* sensu lato and *Alangium* as two distinct families—Cornaceae and Alangiaceae. Xiang (2011), Fu (2019), and Thomas (2021) all supported this conclusion [43,44,45].

As well as the Cornaceae, the subgroups of *Cornus* sensu lato are highly contentious. Some researchers have classified *Cornus* sensu lato into as many as six main genera: *Swida*, *Afrocrania*, *Cornus*, *Cynoxylon*, *Dendrobenthamia* (*Benthamia*), and *Chamaepericlymenum* (*Arctocrania*) [1,8,46,47,48,49,50]. Other scholars advocated dividing *Cornus* sensu lato into 4–10 subgenera [9,26,51,52,53,54,55,56,57,58,59]. For example, Xiang and Boufford (2005) referred to the classification of Ferguson (1966), Xiang (1987), and Murrell (1993), based on cytology, floral anatomy, and chemical substances to indicate that the *Cornus* sensu lato of China were divided into six subgenera: subg. *Yinquania*, subg. *Mesomora*, subg. *Kraniopsis*, subg. *Cornus*, subg. *Syncarpea* and subg. *Arctocrania* [7,54,58,60].

*Cornus* sensu lato could be roughly divided into four groups by the researchers [6,35,58,61,62,63]: (1) the big-bracted dogwoods (BB, subg. *Cynoxylon* and subg. *Syncarpea*); (2) the dwarf dogwoods (DW, subg. *Arctocrania*); (3) the cornelian cherries (CC, subg. *Cornus*, subg. *Sinocornus*, and subg. *Afrocrania*); and (4) the blue- or white-fruited dogwoods (BW, subg. *Kraniopsis*, subg. *Yinquania*, and subg. *Mesomora*). Eyde (1988) described the relationships between taxa as (BW (CC (BB, DW))) based on their morphological characteristics [6]. Some molecular phylogenetic studies supported it [42,45,61,62,63,64,65,66]. Murrell (1993) based on morphological, cytological, phytochemical, and anatomical evidence proposed (BW (DW (CC, BB))) [58]. cDNA sequences of the genes PISTILLATA (PI) and LEAFY (LFY) data analysis are consistent with this conclusion [67,68]. Nowicki (2018) sequenced and analyzed three non-coding cpDNA regions and suggested (BW (CC, BB)) [69]. Thomas (2021) utilized the Angiosperms353 probe set to process phylogenomic data analysis, and the result was (DW (BW (CC, BB))), which differed from all previous hypotheses [45]. Therefore, more efforts are required to disentangle these complex relationships.

Reproductive organs, such as flowers, fruits, and seeds exhibit less morphological variation than vegetative organs. Consequently, they are helpful in plant taxonomy, particularly for distinguishing closely-related species. Xiang (2008) pointed out that the differentiation of Cornaceae fruit colour seems related to the changes in phylogenetic lineages [70]. Woznicka (2014) studied the endocarp of the blue- or white-fruit dogwoods and found that the apical cavity surface morphology, endocarp surface morphology, and the number of vascular bundles on the endocarp surface are taxonomically significant characteristics [71]. Morozowska (2021) mentioned that the endocarp length and thickness, the number of vascular bundles, sclereid shape, and other characteristics were taxonomically essential to *Cornus* sensu lato [72]. Some researchers studied fossils of Cornaceae, but only in a few species [73,74,75,76,77]. However, the research on the fruit of Chinese Cornaceae is limited to just a few subgenera and especially lacks the pericarp structures’ anatomical detail. This study aims to (1) describe more fully the fruit microstructure of Chinese Cornaceae and related taxa; (2) compare the differences in fruit morphology between the genus of Chinese Cornaceae and related taxa and evaluate the relationships among the inner groups of *Cornus* sensu lato; (3) to explore the taxonomical value and importance of fruit features; and (4) provide the fruit morphological evidence for Chinese Cornaceae and related taxa in molecular phylogenetic studies.

## 2. Materials and Methods

The external and internal structures of fruits were recorded from Chinese Cornaceae, representing 30 species and 4 genera following the *Folar of China* [7]. The two species of Araliaceae were also examined [78]. Sample names and voucher information are provided in Table 1.

**Morphological studies**: For each sample, the shape of fruits was first investigated and photographed using an Olympus SZX7 stereomicroscope and an Olympus DP70 digital camera. The fruits were rehydrated, and the epidermis with trichomes was peeled and placed on a glass slide. A drop of 50% glycerol solution was added before mounting the sample with a cover slip. Then, the shape of sclereids and seeds was also observed. At least three mature fruits of each taxon were examined. Photographs were taken using an Olympus BX51 microscope and an Olympus DP70 digital camera.

**Anatomical studies**: Additional rehydrated fruits were placed in FAA (37% formalin–glacial acetic acid and 70% alcohol = 5 mL:5 mL:90 mL) for a minimum of 24 h and then treated following the method of Feder and O’Brien (1968) for embedding in glycol methacrylate (GMA) [79]. A Leica Ultracut R microtome was used to prepare transverse sections about 3–4 μm in thickness. These sections were then stained using the periodic acid Schiff/toluidine blue method. At least three mature fruits of each taxon were examined. In the sections, fruit epidermal cells, mesocarp, endocarp, vascular bundles, secretory structures, and type of crystals were photographed by an Olympus BX51 microscope and an Olympus DP70 digital camera.

The pericarp of fruits was rehydrated and decoloured by sodium hypochlorite and then placed on a glass slide. One or two drops of 50% glycerol solution were added before mounting the sample with a cover slip. The pericarp cells separated from each other by beating the cover slip. These features were observed and photographed using an Olympus BX51 microscope and an Olympus DP70 digital camera and identified similar tannins in cells by ferrous salt [80]. Once more, at least three mature fruits of each taxon were examined.

**Phylogenetic analysis**: Coupled with our original observations of mature fruits and fruit sections, all-important micromorphological and anatomical characters used previously in systematic studies of Chinese Cornaceae and Araliaceae species were examined. Of these, 27 characters were potentially parsimony informative and included in the phylogenetic analysis (Table 2). The data matrix of these characters for the same 32 taxa as examined above for fruit anatomy and micromorphology is presented in Table 3. Araliaceae was used to root the trees. Maximum parsimony (MP) analysis was carried out using PAUP* version 4.0a151 using 1000 random stepwise addition replicate searches and tree bisection and reconnection (TBR) branch swapping [81]. All character states were assumed unordered, and the options multrees, collapse branches, and accurate optimization were selected. Regardless of the number of states, scaling for equal character weighting did not affect the final tree topology. Bootstrap (BS) values were calculated from 1000 to replicate analyses, simple addition sequence of taxa, and TBR branch swapping. All character state changes inferred were mapped onto a single MP tree.

## 3. Results

The morphological and structural characteristics of the fruit were shown in Table 2 and Table 3. The changes in the fruit structure were shown in Figure 1, Figure 2, Figure 3 and Figure 4. The systematic relationships were shown in Figure 5. 

**Trichome:** On the fruits of Cornaceae, except *Helwingia*, lanceolate (e.g., *Aucuba* and *Torricellia*—Figure 2A) and T-shaped (e.g., subg. *Mesomora*, subg. *Cornus*, and subg. *Syncarpea*—Figure 2B) unicellular non-glandular trichomes are found. In subg. *Cornus*, one arm of the T-shaped trichomes is longer than the other (e.g., *Cornus chinensis*—Figure 2C), whereas in other genera the arms are subequal (e.g., *C. controversa*—Figure 2B). As a member of the Araliaceae, *Heteropanax brevipedicellatus* has stellate trichomes (Figure 2D), while *Kalopanax septemlobus* has single-row multicellular trichomes (Figure 2E).

**Pericarp (exocarp, mesocarp, and endocarp):** The exocarp comprises one layer of nearly square parenchymal cells. The outer wall of the exocarp cell is smooth in *Helwingia*, subg. *Arctocrania*, subg. *Cornus*, a portion of subg. *Kraniopsis*, and Araliaceae (e.g., *Cornus alba* and *Kalopanax septemlobus*—Figure 2Q,U), whereas in other taxa, the outer wall of the exocarp cell protrudes outwards (e.g., *Cornus capitata*—Figure 2P).

The mesocarp may consist of 9–30 layers of parenchymal cells. For example, there are 9–10 layers in *Helwingia* (*Helwingia chinensis*—Figure 2M) and about 30 layers in subg. *Syncarpea* (*Cornus capitata*—Figure 2L). The ratio of mesocarp thickness to endocarp thickness is about 5:1 (e.g., *Aucuba himalaica* var. *dolichophylla*—Figure 3C and Figure 4C), 2:1 (e.g., *Cornus capitata*—Figure 3K and Figure 4K), 1:1 (e.g., *C. bretschneideri*—Figure 3R and Figure 4R), or 1:2 (e.g., *C. chinensis*—Figure 3H and Figure 4H). The vascular bundles are located in the mesocarp near the endocarp (e.g., *C. macrophylla*—Figure 2R). Each carpel cross-section has 2–9 vascular bundles (located on the dorsal side of the carpel). In addition, the abdominal vascular bundles are located in the vertical axis of the fruit centre, near the endocarp, on the binding surface between carpels (e.g., *Helwingia japonica*—Figure 4N). The sclereids are located in the mesocarp of subg. *Mesomora*, subg. *Syncarpea*, *Aucuba chinensis* and *Cornus macrophylla* (e.g., *C. macrophylla*—Figure 2R). The secretory cavities are located in the mesocarp of *C. alba*, *C. poliophylla*, and *C. ulotricha* (e.g., *C. ulotricha*—Figure 2S). The mesocarp of the Araliaceae consists of 8–12 layers of parenchyma cells (e.g., *Heteropanax brevipedicellatus*—Figure 2O). The ratio of mesocarp thickness to endocarp thickness is about 3:2 (e.g., *Kalopanax septemlobus*—Figure 3FF and Figure 4FF). Each carpel has five vascular bundles, and the abdominal vascular bundles are parallel to each carpel (e.g., *Kalopanax septemlobus*—Figure 4FF). The oil tubes are located in the mesocarp (e.g., *Kalopanax septemlobus*—Figure 2U).

The endocarp may consist of 2–25 layers of cells. For example, 2–7 layers of cells in *Aucuba* and *Helwingia* (e.g., *Helwingia chinensis*—Figure 2M) and 19–25 layers of cells in subg. *Mesomora*, subg. *Cornus,* and subg. *Syncarpea* (e.g., *Cornus capitata*—Figure 2L). The endocarp is ventrally closed on the carpel, except for *Helwingia* and subg. *Syncarpea* (*Cornus capitata* and *Helwingia chinensis*—Figure 2L,M). The endocarp comprises sclereids (e.g., *Cornus alba*—Figure 2Q). There are three types of sclereid: elongated, with a nearly round cross-section (e.g., *Helwingia japonica*—Figure 2G); polygon, with an irregular cross-section (e.g., *Torricellia angulata*—Figure 2H,T); and nearly round, equal diameter, with a roughly round cross-section (e.g., *Cornus controversa*—Figure 2F). The outer wall of the sclereid is thin, with a ratio of sclereid wall thickness to sclereid width of about 1:6 (e.g., *Torricellia angulata*—Figure 2T), or thick, with a ratio of about 1:3 (e.g., *Cornus alba*—Figure 2Q). The secretory cavities are located in the endocarp of subg. *Cornus* (e.g., *C. chinensis*—Figure 2K). The endocarp of Araliaceae is thin and consists of 4–6 layers of fibres. The endocarp separates at the ventral surface commissure of the carpels (e.g., *Heteropanax brevipedicellatus*—Figure 2O, Figure 3EE and Figure 4EE).

Cornaceae and related taxa fruits contain clusters of calcium oxalate (Figure 2I). The endocarp of subg. *Kraniopsis* contains rhomboidal crystals (Figure 2J), while *Torricellia*, *Helwingia chinensis,* and *H. japonica* lack crystals. The tannins occur in the pericarp (exocarp, mesocarp and endocarp, e.g., *Cornus capitata*—Figure 2P), or not (e.g., *Aucuba japonica*, *Cornus canadensis*, and *Torricellia angulata*). Araliaceae fruits contain clusters of calcium oxalate in mesocarp but no tannins.

**Phylogenetic analysis:** A strict consensus tree was constructed by analyzing the 27 morphological characters using the maximum parsimony method. The MP tree had a length of 110 steps, a consistency index of 0.509, and a retention index of 0.773, accompanying bootstrap support values and distribution of all character state changes indicated in Figure 5. While most branches within Cornaceae and related taxa are weakly supported, four significant clades identified in previous molecular phylogenetic studies are resolved [37,38,39,40,41,42,43,44,45,46,47]: Helwingiaceae (removed *Helwingia* from the Cornaceae), supported by 63% bootstrap support values; Aucubaceae (removed *Aucuba* from the Cornaceae), supported by 93% bootstrap support values; Torricelliaceae (removed *Torricellia* from the Cornaceae); and *Cornus* sensu lato (include subg. *Syncarpea*, subg. *Arctocrania*, subg. *Cornus*, subg. *Kraniopsis*, subg. *Yinquania* and subg. *Mesomora*), supported by 50% bootstrap support values.

## 4. Discussion

Our results indicate that most fruit features can be used as taxonomic evidence to distinguish the genera and subgenera. For example, the difference in the carpel, trichome, the number of cell layers of the mesocarp and endocarp, the ratio of mesocarp thickness to endocarp thickness, the number of vascular bundles, abdominal vascular bundle, secretory structure, sclereid of the mesocarp and endocarp, and cell inclusion. Structural details of these (and other non-fruit) characters and their phylogenetic significance are provided below with reference to the clades identified in the MP analysis (Figure 5) and recent molecular systematic studies of Cornaceae of China and related taxa [35,36,37,38,39,40,41,42,43,44,45]. 

**Helwingiaceae (removed *Helwingia* from the Cornaceae):** Chao (1954) studied the wood anatomy and pollen morphology of nine genera of Cornaceae and concluded that *Helwingia* is located between Cornaceae and Araliaceae [82]. Smith (1975) suggested that *Helwingia* should be promoted to family status because of the distribution of iridoid glycosides, procyanidins, and other compounds [25]. Eyde (1988) recommended that *Helwingia* should be elevated to the family Helwingiaceae based on the flower characteristics (e.g., dioecious, each round of 3–5 tepals, male stamens alternate with tepals) [6]. Wang and Chen (1990) described that the pollen of *Helwingia* lacks a central cavity and a covering layer, compared to the pollen of Araliaceae studied by Shang and Callen (1988) [26,83]. They pointed out that the pollen morphology of *Helwingia* differs from that of Cornaceae and Araliaceae. Noshiro and Baas (1998) proposed that *Helwingia* should be excluded from Cornaceae based on the wood anatomy characters of *Helwingia* (e.g., apotracheal parenchyma absent or rare, fibres septate) [84]. Ao and Tobe (2015), based on the flower and embryology characters (loss of petals, poorly developed disc nectary, tenuinucellate ovules with a mature female gametophyte filled), suggested that *Helwingia* should be raised to family status [29]. Molecular phylogenetic studies supported this conclusion. Morgan and Soltis (1993) determined that *Helwingia* has a distant relationship with Cornaceae based on rbcL sequence analysis [85]. Xiang (1993), according to rbcL sequence data analysis, believed *Helwingia* does not belong to the Cornaceous clade [35]. Based on an analysis of the 18S rDNA sequence, Soltis and Soltis (1997) concluded that there is a distant relationship between *Helwingia* and other genera of Cornaceae [86]. Olmstead (2000) established a phylogenetic tree based on chloroplast ndhF gene sequence data analysis, which supported this conclusion [40]. Savolainen (2000) combined analysis of plastid atpB and rbcL gene sequences [39], and Li (2002) based on rbcL sequence analysis reached the same conclusion [37]. Our research supports previous molecular systematics studies. *Helwingia* has several morphological and anatomical characters of fruit that differentiate it from other species examined herein, including berry drupe (7-0), subglobose (5-4), 3–5 carpels (occasional six carpels) (8-2,3), absence of trichomes (10-0), abdominal vascular bundles close to the endocarp (16-2), an endocarp which is separate at the commissure of carpels (18-0), and an elongated sclereid (22-2). These distinguishing characteristics support that *Helwingia* was separated from other species of Cornaceae.

**Aucubaceae (removed *Aucuba* from the Cornaceae):** Wang and Chen (1990) described that *Aucuba* is not the same as other genera of Cornaceae based on dense drumstick pattern pollen [26]. Noshiro and Baas (1998) noted that the wood anatomy of *Aucuba* (rays up to and over six cells wide, typically in two sizes) differed from other genera of Cornaceae and concluded that *Aucuba* should be excluded from Cornaceae [84]. Molecular evidence confirmed that *Aucuba* has a distant relationship with Cornaceae. Analysis of rbcL sequence data by Xiang (1993) revealed that *Aucuba* is not closely related to the Cornaceae [35]. The rbcL sequences phylogenetic tree established by Morgan and Soltis (1993) proved that *Aucuba* is distantly related to Cornaceae [85]. Savolainen (2000) combined analysis of plastid atpB and rbcL gene sequences and considered that *Aucuba* and *Garrya* are sister branches [39]. Li (2002) determined that *Aucuba* has a distant relationship with *Cornus* sensu lato and *Torricellia* based on rbcL sequence analysis [37]. Ngondya (2013) analyzed cpDNA and ITS sequences and believed that the relationship between *Aucuba* and other Cornaceae genera is distant [38]. Chen (2021) proposed that *Aucuba* is a member of Garryaceae by analyzing plastome DNA sequence data [87]. Huang (2022) indicated that *Aucuba* is monophyletic, a sister branch of *Garrya*, and has a distant relationship with Cornaceae, according to the analysis of 68 plastid protein-coding gene data [88]. Our study supports previous studies and molecular systematics. Members of *Aucuba* have some morphological and anatomical features of fruit that differentiate them from other species studied herein, including berry drupe (7-0), oval (5-1), 10–12 mm (6-3), two carpels (8-0), single-cell lanceolate trichome (10-1), the ratio of mesocarp thickness to endocarp thickness is 4-7 (14-3), vascular bundles number per carpel is 6-9 (15-1), absence of sclereid in mesocarp (17-0), absence of secretory structure (24-0), 2-3 layers of cells in the endocarp (19-0), and nearly-round sclereid (22-1). These distinguishing features separate *Aucuba* from other species of Cornaceae. In determining *Aucuba*’s relationship with Garryaceae, more research is required.

**Torricelliaceae (removed *Torricellia* from the Cornaceae):** Wang and Chen (1990) described that *Torricellia* has acrogenous racemose panicles, which distinguishes it from other Cornaceae groups. As a result, they substantiated its status as an independent family [26]. Noshiro and Baas (1998) pointed out that the wood of *Torricellia* has helical thickenings throughout the vessel elements, which is different from other genera of the Cornaceae, and believed that *Torricellia* should be excluded from this family [84]. Molecular phylogenetic studies provided proof for this conclusion. Savolainen (2000) indicated that *Torricellia* is not closely related to other genera of Cornaceae by using a combined analysis of plastid atpB and rbcL gene sequences [39]. Li (2002) analyzed the rbcL sequences and concluded that *Torricellia* and Cornaceae have a distant genetic relationship [37]. In this study, *Torricellia* displays some morphological and anatomical characteristics of the fruit that set it apart from other species, including drupe (7-1), oval (5-1), three carpels (8-1), single-cell lanceolate trichome (10-1), absence of sclereid in mesocarp (17-0), absence of secretory structure (24-0), sclereid polygon and elongated (22-3), and the absence of crystal and tannin (25-0, 26-0, 27-0). These distinguishing features separate *Torricellia* from other species of Cornaceae.

***Cornus* sensu lato (include subg. *Syncarpea*, subg. *Arctocrania*, subg. *Cornus*, subg. *Kraniopsis*, subg. *Yinquania*, and subg. *Mesomora*):** Murrell (1993) put these subgenera within *Cornus* sensu lato by cladistic analysis of morphological, anatomical, chemical, and cytological characteristics [58]. Noshiro and Baas (1998) studied that the wood anatomy of *Cornus* sensu lato (vessel ray pits, rays that are wider than three cells wide, and typically absent crystals) supported the concept of *Cornus* sensu lato [84]. Previous molecular phylogenetic studies substantiated that *Cornus* sensu lato is a monophyletic taxon and supported the concept of *Cornus* sensu lato [7,35,41,42,43,44,45,61,62,63,64,66]. This group shares several fruit morphological and anatomical characteristics, including drupe (7-1), single-cell T-shaped trichome (10-2), absence of abdominal ventral bundle (16-0), and nearly-round sclereid (22-1). These features distinguish *Cornus* sensu lato from other taxa. Each subgenus has its distinct characteristics: subg. *Arctocrania* fruit has 8–10 layers of cells in the mesocarp (12-0), the ratio of mesocarp thickness to endocarp thickness is about 1:1 (14-1), absence of tannin (27-0); subg. *Cornus* fruit is long and elliptic (5-3), the ratio of mesocarp thickness to endocarp thickness is about 1:2 (14-0), endocarp thickness is 0.5–0.6 mm (20-2), secretory cavities in the endocarp (24-3); subg. *Syncarpea* fruit is an aggregate drupe (7-2), 13–20 mm (6-4), 21–30 layers of cells in the mesocarp (12-3), tannins occur in the exocarp, mesocarp, and endocarp (27-1); subg. *Mesomora* sclereid is in the mesocarp (17-1), the ratio of sclereid wall thickness to sclereid width is about 1:3 (23-2), absence of rhomboidal crystal (25-0); subg. *Yinquania* with four carpels (8-2), tannins occur in the exocarp, mesocarp, and endocarp (27-1); subg. *Kraniopsis* endocarp has 11–15 layers of cells (19-1), and rhomboidal crystal (25-1). Our study supports the previous studies and molecular systematics that the genus *Cornus* sensu lato includes these subgenera.

Adams (1949) described that *Cornus oblonga* has shorter and broader vessel molecules than other species of Cornaceae [89]. Chopra and Kaur (1965) reported that *C. oblonga* produces 1–4 blastocysts, of which only one is fully developed, whereas other species of subg. *Kraniopsis* produce only one blastocyst [90]. Zhu (1984) demonstrated that the pollen grains of *C. oblonga* are oblate, with a slightly pointed sulcus tip and small spiny bulges, which are distinct from *Cornus* sensu lato, resulting in the conclusion that *C. oblonga* should be placed in the separate genus *Yinquania* [91]. Wang and Chen (1990) evaluated that *C. oblonga* should be classified as a separate subgenus subg. *Yinquania* according to pollen with smooth ridges and sparsely short spines [26]. Murrell (1993), according to *C. oblonga* with an oval purplish-red fruit and undisplaced bracts, placed it in Subg. *Yinquania* [58]. By analyzing the rbcL sequences, Xiang (1993) showed that *C. oblonga* occupies an isolated phylogenetic position within *Cornus* sensu lato [35]. Xiang (1998) combined analysis of rbcL-matK sequences and cpDNA restriction site data and identified subg. *Kraniopsis* is sister to subg. *Mesomora*, both of which are the sister group of *C. oblonga* [64]. Yuan (2021) constructed a phylogenetic tree using chloroplast genome sequences and demonstrated that *C. oblonga* is the sister group of other *Cornus* sensu lato species [92]. Our study showed that in addition to four carpels (8-2), four vascular bundles per carpel (15-0), and tannins in the pericarp (27-1), the other characteristics of *C. oblonga* are similar to other species of *Cornus* sensu lato. Thus, we support that *C. oblonga* is classified as Subg. *Yinquania*, and it has a close relationship with subg. *Kraniopsis* in the blue- or white-fruited dogwoods group.

In classical taxonomy studies, Eyde (1988) determined that the relationships among *Cornus* sensu lato as (BW (CC (BB, DW))) based on morphological characteristics [6]. Murrell (1993) proposed (BW (DW (CC, BB))) with morphological, cytological, phytochemical, and anatomical evidence [58]. Wang (2007) believed that subg. *Kraniopsis* is more distantly related to subg. *Cornus*, while with subg. *Mesomora* is closer according to the differences in plant morphology, cell structure, and chemical composition [93]. Morozowska (2021) recommended (BW (CC (DW, BB))) according to morphology, internal structure, and ornamentation of the endocarp [72]. In previous studies of molecular phylogenetics, some studies supported the relationships among *Cornus* sensu lato as (BW (CC (BB, DW))) [42,45,61,62,63,64,65,66], while some studies supported (BW (DW (CC, BB))) [56,67,68,69]. Thomas (2021) revealed a different view with support for (DW (BW (CC, BB))) [45]. This study showed that CC and BB are sister groups with characteristics such as 19–25 layers of cells in the endocarp (19-2). DW is a sister to them. The three groups are bound together by common features, such as crystal clusters in the mesocarp and endocarp (26-3). BW is the base of the clade. In the BW group, subg. *Yinquania* is closely related to subg. *Kraniopsis* in that both have an endocarp with 11–18 layers of cells (19-1) and rhomboidal crystals (25-1). Subg. *Mesomorais* is a sister to the remaining species of this group. Our study supported molecular evidence that the relationships among *Cornus* sensu lato as (BW (DW (CC, BB))) [56,67,68,69].

**Araliaceae 2 genera 2 Species:** Studied species with oblate fruit (5-0), two carpels (8-0), carpel bilateral squashed (9-0), multicellular trichome (10-3), abdominal vascular bundle not close to the endocarp (16-1), the endocarp is composed of fibres (21-0), the endocarp is separate at the commissure of carpels (18-0), and oil tubes located in the mesocarp (24-1). According to the studies of Takhtajan (1969) and Cronquist (1981), Cornaceae should be moved from Apiales to Cornales based on iridoid compounds and that stipules and petioles coalesce into sheaths [4,5]. Wang and Chen (1990) observed that Cornaceae pollen differs from Araliaceae [26]. Our study revealed that Cornaceae and Araliaceae differ more significantly in fruits: (1) trichome—Araliaceae trichomes are stellate or multicellular in a single row, while Cornaceae trichomes are lanceolate or T-shaped; (2) carpel—Araliaceae fruits are bilaterally squashed carpel, while Cornaceae fruits are dorsoventrally squashed carpel; (3) secretory structure—Araliaceae secretory cavity is located in the mesocarp, while Cornus of Cornaceae secretory cavity is located in the endocarp; (4) endocarp—Araliaceae endocarp is composed of fibres, while Cornaceae endocarp is composed of sclereids; and (5) abdominal vascular bundle—Araliaceae has abdominal vascular bundles, while Cornaceae (except *Helwingia*) does not. Our study provides a morphological foundation for previous efforts to remove Cornaceae from Apiales [4,5,94].

## 5. Conclusions

Previous molecular phylogeny studies identified four major branches of Chinese Cornaceae and related taxa studied herein, each of which is now supported by fruit anatomical and micromorphological features. These characteristics are valuable in providing readily observable features for diagnosing monophyletic groups. Characteristics of fruit structure can provide cogent morphological evidence to support the phylogenetic relationships inferred by molecular evidence. They also are useful for predicting the phylogenetic placement of species that have yet to be sampled in molecular studies. However, fruit characteristics on their own cannot wholly resolve species-level relationships, which require combination with additional micromorphological structures of flowers and trophic organs to improve the understanding of this complex group.

## Figures and Tables

**Figure 1 plants-11-02591-f001:**
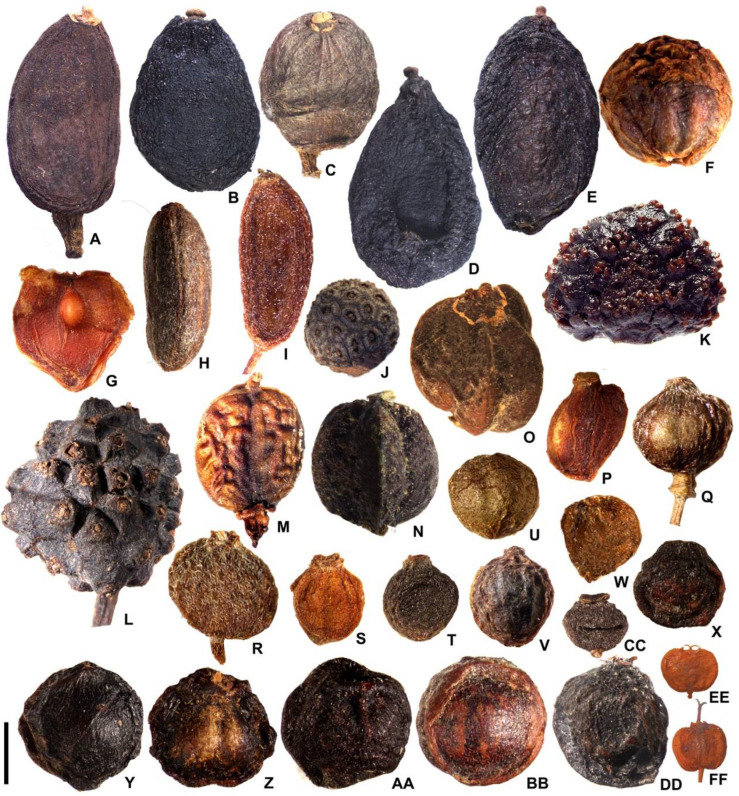
Fruit shape. (**A**) *Aucuba chinensis* var. *angusta*; (**B**) *A. chinensis*; (**C**) *A. himalaica* var. *dolichophylla*; (**D**) *A. japonica*; (**E**) *A. obcordate*; (**F**) *Cornus controversa*; (**G**) *C. canadensis*; (**H**) *C. chinensis*; (**I**) *C. officinalis*; (**J**) *C. elliptica*; (**K**) *C. capitata*; (**L**) *C. hongkongensis*; (**M**) *Helwingia chinensis*; (**N**) *H. japonica*; (**O**) *H. omeiensis*; (**P**) *Cornus alba*; (**Q**) *C. alsophila*; (**R**) *C. bretschneideri*; (**S**) *C. hemsleyi*; (**T**) *C. koehneana*; (**U**) *C. macrophylla*; (**V**) *C. oblonga*; (**W**) *C. quinquenervis*; (**X**) *C. poliophylla*; (**Y**) *C. sanguinea*; (**Z**) *C. schindleri*; (**AA**) *C. ulotricha*; (**BB**) *C. walter*; (**CC**) *C. wilsoniana*; (**DD**) *Torricellia angulate*; (**EE**) *Heteropanax brevipedicellatus*; (**FF**) *Kalopanax septemlobus*; scale bars = 4 mm in (**A**–**I**,**K**–**X**,**CC**); 6 mm in (**J**); and 3 mm in (**Y**–**BB**,**DD**–**FF**).

**Figure 2 plants-11-02591-f002:**
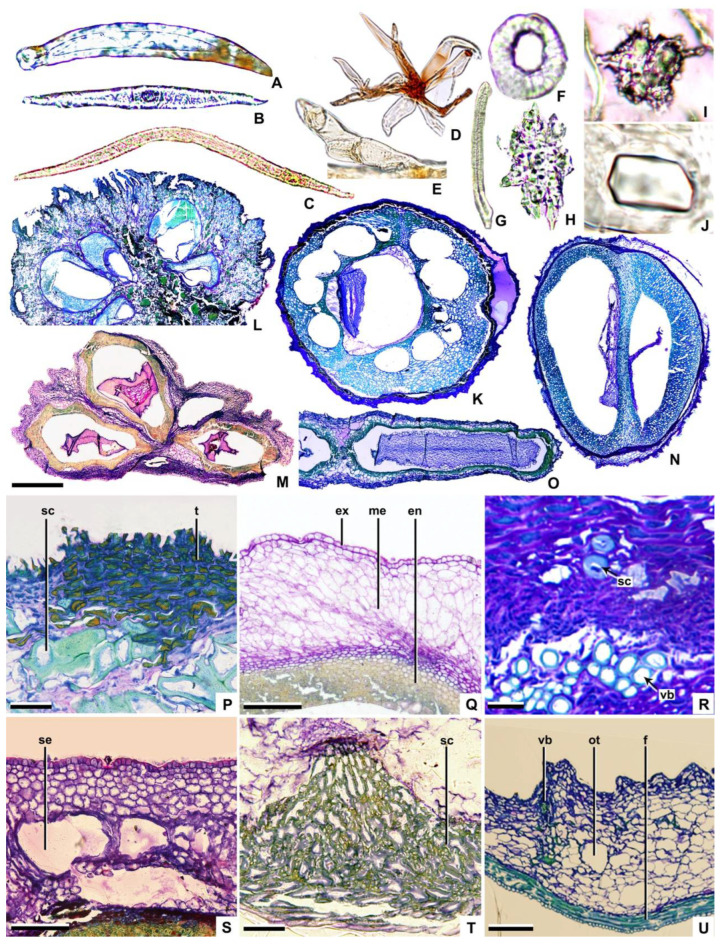
Trichome (**A**–**E**), sclereid (**F**–**H**), crystal (**I**,**J**), transverse sections of fruit (**K**–**U**). (**A**) *Aucuba japonica*; (**B**) *Cornus controversa*; (**C**) *C. chinensis*; (**D**) *Heteropanax brevipedicellatus*; (**E**) *Kalopanax septemlobus*; (**F**) *Cornus controversa*; (**G**) *Helwingia japonica*; (**H**) *Torricellia angulate*; (**I**) *Cornus officinalis*; (**J**) *C. hemsleyi*; (**K**) *C. chinensis*; (**L**) *C. capitata*; (**M**) *Helwingia chinensis*; (**N**) *Cornus macrophylla*; (**O**) *Heteropanax brevipedicellatus*; (**P**) *Cornus capitata*; (**Q**) *C. alba*; (**R**) *C. macrophylla*; (**S**) *C. ulotricha*; (**T**) *Torricellia angulate*; (**U**) *Kalopanax septemlobus*; sc = sclereid, t = tannin, ex = exocarp, me = mesocarp, en = endocarp, vb = vascular bundle, se = secretary cavity, ot = oil tube, f = fiber; scale bars = 50 µm in (**A**,**P**,**T**); 80 µm in (**B**); 100 µm in (**C**,**Q**,**S**); 300 µm in (**D**,**E**); 35 µm in (**F**,**H**,**I**); 70 µm in (**G**); 10 µm in (**J**); 600 µm in (**K**); 400 µm in (**L**,**O**); 800 µm in (**M**); 500 µm in (**N**); 30 µm in (**R**); and 200 µm in (**U**).

**Figure 3 plants-11-02591-f003:**
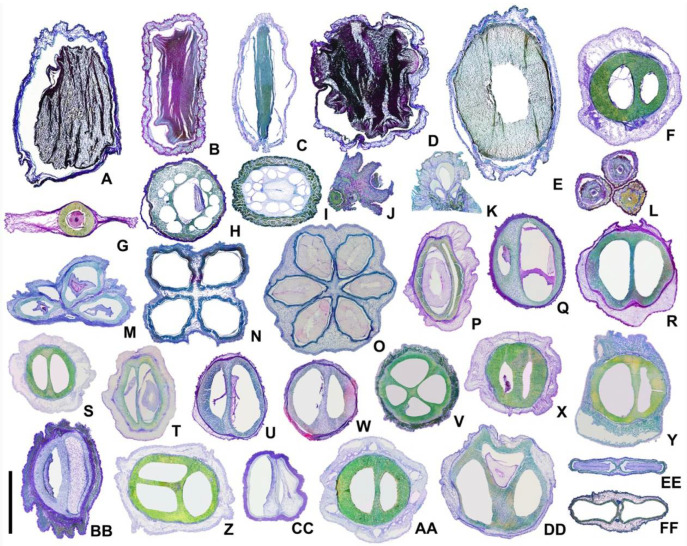
Transverse sections of fruit structures. (**A**) *Aucuba chinensis* var. *angusta*; (**B**) *A. chinensis*; (**C**) *A. himalaica* var. *dolichophylla*; (**D**) *A. japonica*; (**E**) *A. obcordate*; (**F**) *Cornus controversa*; (**G**) *C. canadensis*; (**H**) *C. chinensis*; (**I**) *C. officinalis*; (**J**) *C. elliptica*; (**K**) *C. capitata*; (**L**) *C. hongkongensis*; (**M**) *Helwingia chinensis*; (**N**) *H. japonica*; (**O**) *H. omeiensis*; (**P**) *Cornus alba*; (**Q**) *C. alsophila*; (**R**) *C. bretschneideri*; (**S**) *C. hemsleyi*; (**T**) *C. koehneana*; (**U**) *C. macrophylla*; (**V**) *C. oblonga*; (**W**) *C. quinquenervis*; (**X**) *C. poliophylla*; (**Y**) *C. sanguinea*; (**Z**) *C. schindleri*; (**AA**) *C. ulotricha*; (**BB**) *C. walter*; (**CC**) *C. wilsoniana*; (**DD**) *Torricellia angulate*; (**EE**) *Heteropanax brevipedicellatus*; (**FF**) *Kalopanax septemlobus*; scale bars = 3 mm in (**A**–**I**,**M**–**X**); 2 mm in (**J**–**L**,**EE**,**FF**); and 4 mm in (**Y**–**DD**).

**Figure 4 plants-11-02591-f004:**
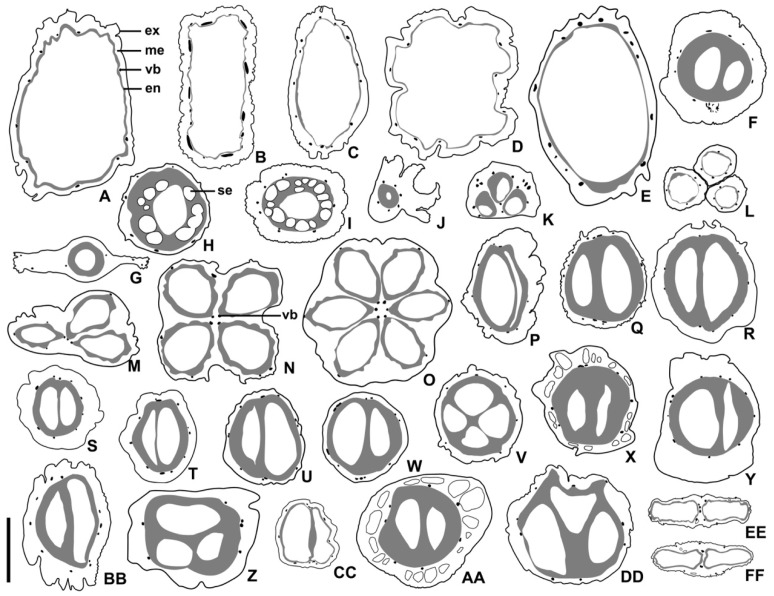
Simplified illustrations of pericarp structures. (**A**) *Aucuba chinensis* var. *angusta*; (**B**) *A. chinensis*; (**C**) *A. himalaica* var. *dolichophylla*; (**D**) *A. japonica*; (**E**) *A. obcordate*; (**F**) *Cornus controversa*; (**G**) *C. canadensis*; (**H**) *C. chinensis*; (**I**) *C. officinalis*; (**J**) *C. elliptica*; (**K**) *C. capitata*; (**L**) *C. hongkongensis*; (**M**) *Helwingia chinensis*; (**N**) *H. japonica*; (**O**) *H. omeiensis*; (**P**) *Cornus alba*; (**Q**) *C. alsophila*; (**R**) *C. bretschneideri*; (**S**) *C. hemsleyi*; (**T**) *C. koehneana*; (**U**) *C. macrophylla*; (**V**) *C. oblonga*; (**W**) *C. quinquenervis*; (**X**) *C. poliophylla*; (**Y**) *C. sanguinea*; (**Z**) *C. schindleri*; (**AA**) *C. ulotricha*; (**BB**) *C. walter*; (**CC**) *C. wilsoniana*; (**DD**) *Torricellia angulate*; (**EE**) *Heteropanax brevipedicellatus*; (**FF**) *Kalopanax septemlobus*; ex = exocarp, me = mesocarp, vb = vascular bundle, en = endocarp, se = secretary cavity; scale bars = 3 mm in (**A**–**I**,**M**–**X**); 2 mm in (**J**–**L**,**EE**,**FF**); and 4 mm in (**Y**–**DD**).

**Figure 5 plants-11-02591-f005:**
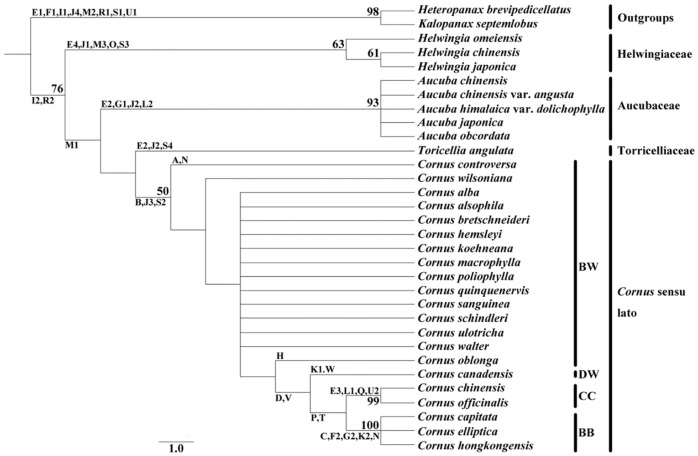
PAUP dendrogram based on fruit characteristics showing the relationships of Araliaceae and Cornaceae, more than 50% of the self-development value is on the branch. (**A**) leaves alternate; (**B**) flowers bisexual; (**C**) umbel; (**D**) phyllary; (**E1**) fruit oblate; (**E2**) fruit oval; (**E3**) fruit long elliptic; (**E4**) fruit subglobose; (**F1**) fruit length 2–3 mm; (**F2**) fruit length 13–25 mm; (**G1**) berry drupe; (**G2**) aggregate drupe; (**H**) three carpels; **(I1**) carpel bilateral squashed; (**I2**) carpel dorsoventral squashed; (**J1**) no trichome; (**J2**) single-cell lanceolate trichome; (**J3**) single-cell T-shaped trichome; (**J4**) multicellular trichome; (**K1**) 8–10 layers of cells in mesocarp; (**K2**) 21–30 layers of cells in mesocarp; (**L1**) the ratio of mesocarp thickness to endocarp thickness is 0.3–0.9; (**L2**) the ratio of mesocarp thickness to endocarp thickness is 3.01–7; (**M1**) no abdominal vascular bundle; (**M2**) abdominal vascular bundle not close to endocarp; (**M3**) abdominal vascular bundle close to endocarp; (**N**) sclereid in mesocarp; (**O**) endocarp is ventrally separate on the carpel; (**P**) 19–25 layers of cells in endocarp; (**Q**) endocarp thickness 0.5–0.6 mm; (**R1**) endocarp composed of fibers; (**R2**) endocarp is composed of sclereid; (**S1**) no sclereid; (**S2**) sclereid nearly round; (**S3**) sclereid elongated; (**S4**) sclereid polygon and elongated; (**T**) the ratio of mesocarp thickness to endocarp thickness is 0.1–0.2; (**U1**) oil tube is located in mesocarp; (**U2**) secretory cavity is located in endocarp; (**V**) mesocarp and endocarp with cluster crystal; (**W**) no tannin.

**Table 1 plants-11-02591-t001:** Araliaceae and Cornaceae taxa were examined for fruit structures and included in the phylogenetic analysis. Species of Cornaceae occurring in China follow *Flora of China*.Herbarium codes for voucher specimens follow Index Herbariorum. Sectional classification of Chinese Cornaceae species follows *Flora of China* [7].

Family	Genus	Subgenus	Species	Voucher Specimens	Locality
Araliaceae	*Heteropanax*		*Heteropanax brevipedicellatus* Li	Lee 200441 (IBSC)	China, Guangdong
	*Kalopanax*		*Kalopanax septemlobus* Thunb.	Wang 1137 (IFP)	China, Liaoning
Aucubaceae	*Aucuba*		*Aucuba chinensis* Benth.	Luo 451 (PE)	China, Hunan
			*A. chinensis* Benth. var. *angusta*	Huang 44350 (PE)	China, Guangdong
			*A.**himalaica* Hoor. var. *dolichophylla*	Dai 106777 (CDBI)	China, Sichuan
			*A. japonica* Thunb.	Shang 6174 (PE)	China, Zhejiang
			*A. obcordata* Rehd.	Y. Tsiang 7512 (NAS)	China, Guizhou
Cornaceae	*Cornus*	*Cornus* subg. *Arctocrania*	*Cornus canadensis* Linn.	Yan 265 (PE)	China, Jilin
		*Cornus* subg. *Cornus*	*C. chinensis* Wanger.	Jie 1615 (CDBI)	China, Sichuan
			*C. officinalis* Sieb.	D.E. Boufford 26065 (PE)	China, Henan
		*Cornus* subg. *Kraniopsis*	*C. alba* Opiz	M.Liu (HANU)	China, Heilongjiang
			*C. alsophila* W.W.Smith	Zhao 0243 (PE)	China, Sichuan
			*C. bretschneideri* Henry	Li 4731 (IFP)	China, Neimenggu
			*C. hemsleyi* Schn.	T.P. Wang 1624 (PE)	China, Shanxi
			*C. koehneana* Wanger.	W.F. Hsia 5277 (PE)	China, Hubei
			*C. macrophylla* Wall.	Harry Smith 6560 (PE)	China, Shanxi
			*C. poliophylla* Schn.	Guan 3063 (PE)	China, Sichuan
			*C. quinquenervis* Farn.	Chen 3109 (CDBI)	China, Chongqing
			*C. sanguinea* Linn.	Zhang 62008 (PE)	China, Beijing
			*C. schindleri* Wanger.	Fang 36766 (PE)	China, Sichuan
			*C. ulotricha* Schn.	Guan 3063 (PE)	China, Sichuan
			*C. walter* Wanger.	Fu 842 (IFP)	China, Jilin
			*C. wilsoniana* Wanger.	Anonymous 2825 (PE)	China, Zhejiang
		*Cornus* subg. *Mesomora*	*C. controversa* Hemsl.	Wang 1506 (PE)	China, Liaoning
		*Cornus* subg. *Syncarpea*	*C. capitata* Wall.	Zhao 4326 (CDBI)	China, Sichuan
			*C. elliptica* Pojark.	Mei 793 (AU)	China, Fujian
			*C. hongkongensis* Hemsl.	Mei 116 (AU)	China, Fujian
		*Cornus* subg. *Yinquania*	*C. oblonga* Wall.	Xi 12122 (CDBI)	China, Sichuan
Helwingiaceae	*Helwingia*		*Helwingia chinensis* Batal.	Zhi 8237 (CDBI)	China, Sichuan
			*H. japonica* Thunb.	Song 1084 (CDBI)	China, Sichuan
			*H. omeiensis* Fang	Yang 34 (CDBI)	China, Sichuan
Torricelliaceae	*Torricellia*		*Torricellia angulata* Oliv.	Yang 292 (CDBI)	China, Sichuan

**Table 2 plants-11-02591-t002:** Characters and character states used in the phylogenetic analysis of fruit anatomical and micromorphological characters in Chinese Cornaceae and related taxa.

Character No.	Character	States
1	Leaf growth pattern	0 = Alternate; 1 = Opposite
2	Flower sexuality	0 = Unisexuality; 1 = Bisexuality
3	Inflorescence	0 = Panicle; 1 = Cyme; 3 = Umbel; 4 = Capitulum
4	Phyllary	0 = Absent; 1 = Present
5	Fruit shape	0 = Oblate; 1 = Oval; 2 = Sphericity; 3 = Long elliptic; 4 = Subglobose
6	Fruit length (mm)	0 = 2–3; 1 = 4–6; 2 = 7–9; 3 = 10–12; 4 = 13–25
7	Fruit type	0 = Berry drupe; 1 = Drupe; 2 = Aggregate drupe
8	Carpel number	0 = 2; 1 = 3; 2 = 4–6
9	Carpel shape	0 = Bilateral squashed; 1 = Dorsoventral squashed
10	Trichome	0 = None; 1 = Single-cell lanceolate; 2 = Single-cell T-shaped; 3 = Multicellular
11	Outer wall of epidermal cell	0 = Flattened; 1 = Bulging
12	Cell layer of mesocarp	0 = 8–10; 1 = 11–15; 2 = 16–20; 3 = 21–30
13	Mesocarp thickness (mm)	0 = 0.1–0.3; 1 = 0.31–0.8
14	Ratio of mesocarp thickness to endocarp thickness	0 = 0.3–0.9; 1 = 0.91–1.1; 2 = 1.11–3; 3 = 3.01–7
15	Vascular bundle per carpel	0 = 3–5; 1 = 6–8
16	Abdominal vascular bundle	0 = Absent; 1 = Present, not close to endocarp; 2 = Present, close to endocarp
17	Sclereid in mesocarp	0 = Absent; 1 = Present
18	Endocarp at commissure of carpels	0 = Separate; 1 = Combined
19	Cell layer of endocarp	0 = 2–10; 1 = 11–18; 2 = 19–25
20	Endocarp thickness (mm)	0 = 0.03–0.1; 1 = 0.2–0.4; 2 = 0.5–0.6
21	Endocarp constitute	0 = Fiber; 1 = Sclereid
22	Sclereid shape	0 = None; 1 = Nearly round; 2 = Elongated; 3 = Polygon and elongated
23	Ratio of sclereid wall thickness to sclereid width	0 = None; 1 = 0.1–0.2; 2 = 0.21–0.4
24	Secretory structure	0 = None; 1 = Oil tube; 2 = Secretory cavity is located in mesocarp; 3 = Secretory cavity is located in endocarp
25	Rhomboidal crystal	0 = Absent; 1 = Present
26	Cluster crystal	0 = Absent; 1 = Mesocarp; 2 = Endocarp; 3 = Mesocarp and endocarp
27	Tannin	0 = None; 1 = Exocarp, mesocarp and endocarp; 2 = Exocarp and mesocarp; 3 = Mesocarp and endocarp; 4 = Mesocarp

**Table 3 plants-11-02591-t003:** Data matrix of fruit characters used in the phylogenetic analysis of Chinese Cornaceae species and related taxa. Characters and states are described in Table 2. Characters 1–4 from *Flora of China* [7].

Species	Character																										
	1	2	3	4	5	6	7	8	9	10	11	12	13	14	15	16	17	18	19	20	21	22	23	24	25	26	27
*Heteropanax brevipedicellatus*	0	1	0	0	0	0	0	0	0	3	0	0	0	2	0	1	0	0	0	0	0	0	0	1	0	1	0
*Kalopanax septemlobus*	0	1	0	0	0	0	0	0	0	3	0	1	0	2	0	1	0	0	0	0	0	0	0	1	0	1	0
*Aucuba chinensis*	1	0	0	0	1	3	0	0	1	1	1	1	0	3	1	0	1	1	0	0	1	1	1	0	0	1	4
*A. chinensis* var. *angusta*	1	0	0	0	1	3	0	0	1	1	1	1	0	3	1	0	0	1	0	0	1	1	1	0	0	1	4
*A.**himalaica* var. *dolichophylla*	1	0	0	0	1	3	0	0	1	1	1	1	0	3	1	0	0	1	0	0	1	1	1	0	0	1	2
*A. japonica*	1	0	0	0	1	3	0	0	1	1	1	1	0	3	1	0	0	1	0	0	1	1	1	0	0	1	0
*A. obcordata*	1	0	0	0	1	3	0	0	1	1	1	1	0	3	1	0	0	1	0	0	1	1	1	0	0	1	2
*Cornus canadensis*	1	1	1	1	2	1	1	0	1	2	0	0	0	1	0	0	0	1	1	1	1	1	2	0	0	3	0
*C. chinensis*	1	1	2	1	3	2	1	0	1	2	0	0	0	0	0	0	0	1	2	2	1	1	1	3	0	3	4
*C. officinalis*	1	1	2	1	3	2	1	0	1	2	0	2	0	0	0	0	0	1	2	2	1	1	1	3	0	3	4
*C. alba*	1	1	1	0	2	1	1	0	1	2	0	1	1	1	0	0	0	1	1	1	1	1	2	2	1	2	3
*C. alsophila*	1	1	1	0	2	1	1	0	1	2	1	1	1	0	1	0	0	1	1	1	1	1	2	0	1	2	3
*C. bretschneideri*	1	1	1	0	2	1	1	0	1	2	1	1	1	1	0	0	0	1	1	1	1	1	2	0	1	2	3
*C. hemsleyi*	1	1	1	0	2	1	1	0	1	2	0	1	1	1	0	0	0	1	1	1	1	1	2	0	1	2	4
*C. koehneana*	1	1	1	0	2	1	1	0	1	2	1	1	1	2	1	0	0	1	1	1	1	1	2	0	1	2	3
*C. macrophylla*	1	1	1	0	2	1	1	0	1	2	1	1	1	1	1	0	1	1	1	1	1	1	2	0	1	2	3
*C. poliophylla*	1	1	1	0	2	1	1	0	1	2	0	1	1	1	0	0	0	1	1	1	1	1	2	2	1	2	3
*C. quinquenervis*	1	1	1	0	2	1	1	0	1	2	1	1	1	1	1	0	0	1	1	1	1	1	2	0	1	2	3
*C. sanguinea*	1	1	1	0	2	1	1	0	1	2	1	2	1	2	1	0	0	1	1	1	1	1	2	0	1	2	3
*C. schindleri*	1	1	1	0	2	1	1	1	1	2	0	1	1	1	0	0	0	1	1	1	1	1	2	0	1	2	3
*C. ulotricha*	1	1	1	0	2	1	1	0	1	2	1	1	1	1	0	0	0	1	1	1	1	1	2	2	1	2	3
*C. walter*	1	1	1	0	2	1	1	0	1	2	1	2	1	1	1	0	0	1	1	1	1	1	2	0	1	2	3
*C. wilsoniana*	1	1	1	0	2	1	1	0	1	2	1	1	0	1	1	0	0	1	1	1	1	1	1	0	1	2	3
*C. controversa*	0	1	1	0	2	1	1	0	1	2	1	1	0	1	1	0	1	1	2	1	1	1	2	0	0	2	3
*C. capitata*	1	1	3	1	2	4	2	0	1	2	1	3	1	2	0	0	1	0	2	1	1	1	1	0	0	3	1
*C. elliptica*	1	1	3	1	2	4	2	0	1	2	1	3	1	3	0	0	1	0	2	1	1	1	1	0	0	3	1
*C. hongkongensis*	1	1	3	1	2	4	2	0	1	2	1	3	1	2	0	0	1	0	2	1	1	1	1	0	0	3	1
*C. oblonga*	1	1	1	0	2	1	1	2	1	2	0	2	0	2	0	0	0	1	1	1	1	1	2	0	1	2	1
*Helwingia chinensis*	0	0	2	0	4	2	0	1	1	0	0	0	0	1	0	2	0	0	0	1	1	2	2	0	0	0	4
*H. japonica*	0	0	2	0	4	2	0	2	1	0	0	0	0	1	0	2	0	0	0	0	1	2	2	0	0	0	4
*H. omeiensis*	0	0	2	0	4	2	0	2	1	0	0	0	0	2	0	2	0	0	0	0	1	2	2	0	0	1	0
*Torricellia angulata*	0	0	0	0	1	1	1	1	1	1	1	1	0	1	0	0	0	1	1	1	1	3	1	0	0	0	0

## Data Availability

Not applicable.

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
