# Peer review of "The Phylogenetic Significance of Fruit Structures in the Family Cornaceae of China and Related Taxa"

_plants, 2022, doi:10.3390/plants11192591_

Round 1

Reviewer 1 Report

This manuscript offers a review on the systematics of Cornales and the history of its treatments. It is very well synthesized and the taxonomic treatments for Cornus are also well presented. Because there is no consensus for the phylogenetic placement of species within the genus, the authors search for anatomical features of the fruit.

The major contribution of this research is related to the description of fruit characters within Cornales. The phylogenetic approach is welcome to understand their possible evolution within the group. However, to truly comprehend the evolution of the fruit the inclusion of molecular sequence data must go along with the morphological data.

Line 256-257. The phylogenetic strict consensus tree, based on 27 morphological characters, showed Cornus as monophyletic but with no support at all. It would be wrong to assume 50% as BS, as it means that in 50% of the trees the clade is recovered but in the other 50% is not.  Hence, its alliance with the other families is moderately supported with the 76 BS value.

Decisions to include or exclude different genera from Cornaceae are based on a poorly supported morphological phylogeny. Thus, although a review of previous treatments is presented, the evidence to retain any conclusion is weak.

Characteristics of fruit structure can be useful to set infrageneric categories. However, it would be necessary to include other characters (molecular/morphological ones) to resolve relationships within Cornus and Cornales.

I encourage the authors to use morphological characters as descriptors of species/sections and to combine them with molecular data in a robust phylogeny to decide on taxonomic changes and relationships.

Author Response

Dear Reviewer:

We sincerely appreciate your thoughtful comments on our manuscript entitled “The Phylogenetic Significance of Fruit Structures in the Family Cornaceae of China and Related Taxa” (ID: 1870574). Those comments are very helpful for improving our paper.

We would like to thank you for pointing out the problem that we should use morphological characters as descriptors of species/sections and to combine them with molecular data in a robust phylogeny to decide on taxonomic changes and relationships.

We have revised the entire text, marked up using the “Track Changes” function. We have combined our fruit morphology with existing molecular phylogenetic classification systems to compare differences in fruit structure between taxa, explore the taxonomical value and importance of fruit features, provide the fruit morphological evidence for molecular phylogenetic studies, and understand their possible evolution within the group.

Thanks for pointing out the poor support for Cornus as a monophyletic taxon. We made a change to refer to previous studies in molecular systematics on how to determine clades (Lines 309-311).

Once again, thank you very much for your recognition and suggestion. It will substantially improve our research.

With best wishes,

Sincerely,

Jingru Wang

Reviewer 2 Report

Please strengthen the "conclusions" paragraph; if fruit characteristics alone are insufficient to resolve interspecific issues, then write another statement on what can help further resolve these issues.

1. As for the methodology - I noted that three mature fruits from each taxon was used for the study. I am not sure if more samples should have been studied.

2. If previous studies have employed molecular techniques to separate several taxa from Cornaceae, isn't this paper merely proving the molecular analysis of other authors?

3. There should have been a "stronger" or more convincing statement as to the significance of the study.

Author Response

Dear Reviewer:

We sincerely appreciate your thoughtful comments on our manuscript entitled “The Phylogenetic Significance of Fruit Structures in the Family Cornaceae of China and Related Taxa” (ID: 1870574). Those comments are very helpful for improving our paper.

We would like to thank you for pointing out the problem that we should strengthen the "conclusions" paragraph; if fruit characteristics alone are insufficient to resolve interspecific issues, then write another statement on what can help further resolve these issues.

Reply: On lines 570-573 of the "conclusions" we added what can help further resolve these issues, which is a vision for future work.

  1. As for the methodology - I noted that three mature fruits from each taxon was used for the study. I am not sure if more samples should have been studied.

Reply: Thanks for your comments. We refer to the previous studies and consider it feasible.

Liu, M., Van Wyk, B-E., Tilney, P.M., et al. The phylogenetic significance of fruit structural variation in the tribe Heteromorpheae (Apiaceae). Pak. J. Bot., 2016, 48(1): 201-210.

Liu, M., Downie, S.R. The phylogenetic significance of fruit anatomical and micromorphological structures in Chinese Heracleum species and related taxa (Apiaceae). Systematic Botany, 2017, 42(2): 313–325.

  1. If previous studies have employed molecular techniques to separate several taxa from Cornaceae, isn't this paper merely proving the molecular analysis of other authors?

Reply: We are thankful to the reviewer for his critical comments and regret the confusing description of the logic. We think this issue, like the third issue you raised, is a problem caused by our not getting to the point in the sense of the article. We should combine our fruit morphology with existing molecular phylogenetic classification systems to compare differences in fruit structure between taxa, explore the taxonomical value and importance of fruit features, provide the fruit morphological evidence for existing molecular phylogenetic studies, to understand their possible evolution within the group. We have revised the entire text, marked up using the “Track Changes” function.

  1. There should have been a "stronger" or more convincing statement as to the significance of the study.

Reply: The same answer as the previous question, we have been modified in the whole text, marked up using the "Track Changes" function.

Once again, thanks for your suggestion. It’s a critical comment that will substantially improve our research. 

With best wishes,

Sincerely,

Jingru Wang

Round 2

Reviewer 1 Report

Thank you for clarifying the suggestions made in the previous version. 

Author Response

Thanks very much for your recognition and suggestion. With our best wishes.